# Agronomic, Sensory and Essential Oil Characterization of Basil (*Ocimum basilicum* L.) Accessions

Fernanda Abduche Galvão Pimentel [1,*], Mariana Altenhofen da Silva [2],
Simone Daniela Sartorio de Medeiros [3], José Magno Queiroz Luz [4] and Fernando César Sala [5]

1   Araras Campus, Federal University of São Carlos (UFSCar), São Paulo 13604-900, Brazil
2   Department of Agroindustrial Technology and Rural Socioeconomics, Araras Campus,
    Federal University of São Carlos (UFSCar), São Paulo 13604-900, Brazil; mari.altenhofen@gmail.com
3   Department of Informatics and Statistics, Trindade Campus,
    Federal University of Santa Catarina (INE/CTC, UFSC), Santa Catariana 88040-900, Brazil;
    sisartorio@gmail.com
4   Glória Campus, Federal University of Uberlândia (UFU), Uberlândia 38400-000, Brazil; jmagno@ufu.br
5   Department of Biotechnology and Plant and Animal Production, Araras Campus,
    Federal University of São Carlos (UFSCar), São Paulo 13604-900, Brazil; fcsala@ufscar.br
*   Correspondence: fernandaagp@estudante.ufscar.br

**Abstract:** Basil (*Ocimum basilicum* L.) is one of the main condiments for fresh consumption and essential oil production. The aim of the present work was to assess the agronomic characterization and analyze the essential oil of 63 basil accessions. The experiment was conducted in two stages in a greenhouse using vases and a hydroponic cultivation system. Oil extraction was performed employing the Soxhlet method. There was a significant variation in the agronomic characteristics among the evaluated accessions. The estimated total contents of essential oils ranged from 0.05 to 0.40%, and the major volatile fraction was methyl-eugenol. In the present study, accessions with superior performance compared to commercial varieties were found. Accession BL11 presented agronomic characteristics suitable for cultivation in a hydroponic system due to its better plant structure and late flowering. Accession BL24 stood out for essential oil extraction, producing 17.6% of linalool and a high intensity of color and odor. Accessions BL11 and BL24 presented market potential, given their higher mass productivity and higher essential oil yield, respectively. These accessions can be made available as new varieties in addition to being used in genetic improvement programs for this species.

**Keywords:** linalool; early flowering; hydroponic system; essential oil content and composition; sensory analysis

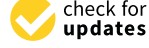



## 1. Introduction

Basil (*Ocimum basilicum* L.) is a plant belonging to the family *Laminaceae*, originating from North India [1], with diverse purposes of use in the world market [2]. In Brazil, basil has been introduced by the Italians, being mainly cultivated in the State of Sergipe [3], with emphasis for cultivation in family agriculture [4]. It presents a great polymorphism and, therefore, exhibits different agronomic characteristics regarding plant height, leaf color and inflorescence, besides the differentiation in the cultivation system, aroma and contents of essential oils [5,6].

The difficulty in characterizing the *O. basilicum* varieties probably derives from the occurrence of cross-pollination, resulting in a large number of subspecies, varieties and shapes. Thus, to assist the research works on the genetic improvement of this species, it is important to perform the characterization of the accessions to be used. The evaluation and characterization of varieties belonging to germplasm collections may enable their exploration and the use of their prominent characteristics, with the preservation of these

species by seeds [7,8]. Therefore, to assist in the research on the genetic improvement of these species, it is important to characterize the accessions to be used.

In previous works, the characterization of varieties of *O. basilicum* was carried out, in which 17 accessions of this species cultivated in a greenhouse were [9] evaluated. After 60 days of cultivation, different agronomic characteristics were evaluated. Observing that the "small" type of basil was characterized by small size, a thicket habit, small leaves and a smooth leaf margin, the narrow leaves genotype was the one that presented the highest plant height and serrated leaf margin. While some evaluated accessions presented an elliptical leaf blade, also reporting that the size of the petiole can vary according to the age of the leaf, in large leaves, however, petioles tend to be naturally larger than in small ones. It should be noted that this article is in line with our results, demonstrating the great diversity and importance of characterizing types in *O. basilicum*.

Basil varieties with late flowering are ideal for the commercialization of the plant *in natura*, since there is a greater interest in the leaves and not in inflorescence. Another greatly explored fraction of basil, extracted from the leaves, is essential oil, whose main constituents are linalool, methyl chavicol, geranial, eugenol and methyl eugenol, found among the different basil chemotypes [10]. Nevertheless, around 36 chemical constituents can be identified in the composition of the essential oils of this species [11].

According to [12], the two heights of the cuts applied at the time of harvesting, which were carried out at three different times throughout the development of the crop, influenced the essential oil yield, noting that the essential oil contents in the basil cultivars ranged from 1.72 to 0.32%. In addition, they found that in the third season, the oil yield was 37.14% lower compared to the averages of the first two harvest seasons. Therefore, the older the plant was, the lower its oil yields were, highlighting that all these evaluated plants were in the vegetative stage.

In previous studies, the chemical composition of the essential oil of basil was observed, where 52 compounds were identified, with the major component being linalool (41.3%) followed by 1,8-cineol (9.6%), (Z)-isoeugenol (5.9%), 1-epi-cubenol (4.8%), $\alpha$-transbergamotene (4.6%) and (Z)-anethole (3.2%). Other compounds occur in amounts between 2 and 3%, and finally, the others are present in amounts below 2% [13].

Essential oils are active principles activated by the secondary metabolism as a consequence of some stress suffered by the plant [14], common in crops in the conventional system. The biosynthesis of essential oils can be influenced by several factors, such as plant development stage and climatic factors related to temperature, humidity, precipitation, solar radiation and photoperiod [15], whereas the chemical composition of essential oils is determined by genetic factors, development stage, plant age and the interactions among the plant, microorganisms and insects [16]. Due to this diversity, it is important to know the content and composition of the essential oil of the basil variety worked.

Another potential use of basil essential oil is as antimicrobial, antiparasitic and medicinal, which has driven its demand in the international market [17,18]. Given this diversity of use and benefits, basil has been gaining prominence in the economic scene. Therefore, new production systems have been implemented with a growth trend for cultivation in a protected environment employing the hydroponic system [19].

Given the above, the present work aimed at performing the agronomic characterization of the content and composition of essential oil in new basil accessions not yet characterized.

## 2. Material and Methods

The experiments were performed in the experimental area of the Horticulture sector of the Department of Biotechnology and Plant and Animal Production (DBPVA), of the Center for Agrarian Sciences (CCA), of the Federal University of São Carlos (UFSCar), Araras Campus/SP, in the period from 27 June to 10 December 2017 (1st step) and from 15 December 2018 to 20 February 2019 (2nd step).

### 2.1. First Step: Agronomic Characterization of Basil Accessions

2.1.1. Accessions Used and Seedling Acquisition

A total of 63 basil accessions from the Germplasm Bank of Basil of the Federal University of São Carlos–Campus Araras city (São Paulo, Brazil) were used. The seedlings were produced in 128-cell trays filled with coconut-fiber substrate, and after 30 days, they were transplanted to vases (1 L) with the substrate Spagnhol® and conducted in a protected environment and in a fertigation system. Each accession was represented by one plant.

2.1.2. Agronomic Characterization

After transplanting, each accession was evaluated weekly regarding the beginning of flowering, classifying them into the following types: early (less than 40% of flowers within the first three weeks of evaluation), intermediary (less than 40% of flowers among the fourth, fifth and sixth weeks of evaluation) and late (beginning of flowering from the seventh week of evaluation). The plants that flowered were analyzed regarding flowering percentage.

The flowering percentage of the plant was determined by the weekly count of the number of inflorescences per plant until the end of the cycle with the following scale: 0–20% (presence of little inflorescence), 20–40% (almost half of the plant with the presence of inflorescence), 40–60% (at least half of the plant with the presence of inflorescence), 60–80% (more than half with the presence of inflorescence) and 80–100% (the whole plant with inflorescence).

After 100 days of cultivation, the following characteristics were evaluated: plant height (cm); plant diameter (cm); width and length of the leaf blade of the most developed leaf (cm)—the plants with the leaf blade above 1.5 cm in width were classified as broad leaf (BL) accessions, whereas narrow leaves (NL) were below this value; number of primary branches obtained with the number of branches developed from the main stem; internode length—distance (cm) between internodes on the same (main) branch at the median part of the plant; length of the branch with inflorescence (cm); percentage of seed maturation in the inflorescence—a scale of scores of 0–20% (up to 20% of ripe seeds), 21–40% (up to 40% of ripe seeds), 41–60% (up to 60% of ripe seeds), 61–80% (up to 80% of ripe seeds) and 81–100% (>81 up to 100% of ripe seeds) was employed; weight of the produced seeds—the plants were harvested individually, stored in paper bags and dried to obtain seed weight. For the second stage, 13 accessions classified among the intermediate and late flowering groups were used, obtaining viable seeds.

### 2.2. Second Step: Agronomic Characterization of the Plants and Analysis of the Essential Oil

In the 2nd step, the accessions were conducted in a hydroponic cultivation system of the NFT (laminar flow of nutrients) type.

The following accessions were evaluated: NL12, NL13, NL16, NL19, NL26, NL28 (intermediary flowering) and BL11, BL24, BL22, NL29, NL34, NL35, NL38 (late flowering). The commercial varieties Manolo (Broad Leaf type) and Fino Francês (Narrow Leaf type) were used as controls.

The seedlings were produced in trays of 128 cells and, after 30 days, transplanted and cultivated in profiles of 50 mm in diameter in a hydroponic system of the NFT type, in an environment protected with a red ChromatiNet® Leno 20% mesh.

The experiment was conducted in randomized blocks with plots containing 15 plants with a spacing of 0.20 m between plants and profiles and with 3 repetitions. Three central plants from each plot were used for evaluation, excluding those of the border.

The nutrient solution employed was [20] 120 g 1000 L$^{-1}$ of MAP (monoammonium phosphate, N: 11% + P$_2$O$_5$: 60%, brand Ominia®); 500 g 1000 L$^{-1}$ of calcium nitrate (N: 15.5% + Ca: 19%, brand YaraLiva®); 650 g 1000 L$^{-1}$ of potassium nitrate (N: 12% + K$_2$O: 45%, S: 1.2%, brand DripSol®); 350 g 1000 L$^{-1}$ of magnesium sulfate (Mg: 9% + S: 11.9%, brand Heringer®); 20 g 1000 L$^{-1}$ of a micronutrient cocktail (B: 1.82%-Cu EDTA: 1.82%-Fe EDTA: 7.26%-Mn EDTA: 1.82%-Mo:0.3%, Ni: 0.335%-Zn EDTA: 0.73%, brand Conplant®); and 30 g 1000 L$^{-1}$ of FeQ48-iron chelate (Fe: 16%, brand DripSol®).

Daily measurements of electrical conductivity were performed in the culture solution, which was maintained at 1.4 μS cm$^{-1}$, and of pH, maintained between 5.5 and 6.5.

### 2.2.1. Agronomic Characterization

After 60 days of transplanting, the following parameters were evaluated: diameter of the aerial part (cm); plant height (cm); width and length (cm) of the leaf blade of the most developed leaf; fresh mass of the aerial part (g); fresh mass of the root (g); number of primary branches; internode length (cm) measured in the middle part of the plant; and root length (cm).

### 2.2.2. Sensory Evaluation

After 60 days of transplanting, the following sensory characteristics were evaluated: plant coloration intensity from a scale of green color intensity: low (score 0), intermediate (score 1) and high (score 2), and odor intensity, classifying them into score 1 (absence of odor), score 2 (little odor), score 3 (intermediate odor), score 4 (high odor) and score 5 (very high odor).

### 2.2.3. Extraction and Analysis of the Essential Oil

The essential oil content in the accessions was determined through extraction with a solvent (Soxhlet) following the methodology of [2], with some modifications, since a lower amount of the solvent was used, and it was recovered in the process.

A composite sample, containing only the leaves (three central plants) was used and subsequently dried in an air circulation oven at 45 °C for 48 h and ground. Aliquots of 5 g of sample were stored in paper bags (in triplicate), which were placed in individual reboiler tubes previously washed and dried. Inside each tube, 150 mL of cyclohexane (solvent used in the extraction process) were added. The samples remained in contact with the solvent at reflux for 2.5 h at 115 °C. Subsequently, the bags were suspended for solvent recovery and sample washing. The lipid fractions were transferred to 25 mL flasks with the solvent.

The oil samples were analyzed in a Shimadzu GC-2010 gas chromatograph with AOC-20i automatic auto-injector, capillary column RTX-Wax™ (60 m, 0.25 mm ID, 0.25 μm df, Restek©) and flame ionization detector (FID). The identification of compounds in oil samples was performed by comparing their retention times with those of pure standards of Linalool (Ref. W263508, Sigma Aldrich, St. Louis, MO, USA), Methyl Eugenol (Ref. W247502, Sigma Aldrich, St. Louis, MO, USA) and Eugenol (Ref. W266700, Sigma Aldrich, St. Louis, MO, USA) [21]. The content of essential oil in the samples was also estimated.

An aliquot of 1 μL of sample was injected with a splitting ratio of 40:1, using helium as drag gas at the linear speed of 22 cm.s$^{-1}$ and obtaining the separation of the analytes in a chromatographic run of 25 min. The temperatures of the injector and detector were, respectively, 230 °C and 300 °C, and the initial temperature of the column was 100 °C.

The programming of the temperature of the oven that housed the column was from 100 to 160 °C at the speed of 10 °C min$^{-1}$, residence time of 1 min; 160 to 178 °C at the speed of 10 °C min$^{-1}$, residence time of 1 min; 178 to 212 °C at the speed of 7 °C min$^{-1}$, residence time of 1 min; 212 to 228 °C at the speed of 3 °C min$^{-1}$, without residence time; 228 to 240 °C at the speed of 120 °C min$^{-1}$, residence time of 4 min.

The determination and integration of the peaks were performed using the software GCsolution v. 2.42.00 (Shimadzu©). To determine the estimated content of essential oil in the samples, Equation (1) was employed.

$$E.O.\ Content = \frac{\sum C_{sq}}{\sum PR_{sq} \times C_{me}} \times 100 \tag{1}$$

where "*E.O. Content*: content of essential oil in the sample (%)", "$\sum C_{sq}$: sum of the concentrations of the identified and quantified substances (mg/L)", "$\sum PR_{sq}$: sum of the relative percentages of the identified substances in terms of peak area in relation to the total area of

the peaks observed in the run (%)", "$C_{me}$: concentration of the wet mass extracted from the plant material in relation to the final volume of extract, after centrifugation (mg/L)".

### 2.2.4. Statistical Analysis

The data on the agronomic characterization of the accessions (1st and 2nd steps), oil content and the volatile fractions of linalool, methyl eugenol and eugenol were analyzed by the analysis of variance (ANOVA). In the 1st step, a completely randomized design in a $2 \times 3$ factorial scheme was used, whereas in the 2nd step, it was a completely randomized design. When necessary, Tukey's test was applied along with the principal component analysis for the variables analyzed at the end of the experiment, considering the mean value of the variables in each accession. All analyses were performed using the software R, considering a significance level of 5%.

## 3. Results and Discussion

### 3.1. First Step: Agronomic Characterization of Basil Accessions

There were different flowering periods among the evaluated accessions (Table 1). Late flowering is desirable for the consumer market of basil plants, since the main commercialization is *in natura* to be used as a condiment.

**Table 1.** Classification of the accessions in the flowering groups according to the flowering period evaluated over time.

| Accessions | Flowering Period |
|---|---|
| BL4, BL5, BL6, BL7, BL8, BL9, BL10, BL12, BL13, BL14, BL17, BL18, BL19, BL20, BL21, BL23, BL25, NL1, NL2, NL3, NL7, NL8, NL9, NL10, NL14, NL17, NL21, NL23, NL24, NL25 | Early |
| BL1, BL2, BL3, BL15, BL16, BL4, BL5, BL6, BL11, NL12, NL13, NL15, NL16, NL18, NL19, NL20, NL22, NL26, NL27, NL28, NL30, NL31, NL36 | Intermediary |
| BL24, BL22, NL29, NL32, NL33, NL34, NL35, NL37, NL38 | Late |

Plants with early flowering tend to decrease the production of the number of leaves, harming their commercialization *in natura*. The removal of the first inflorescences is recommended to increase the number of leaves and the vegetative cycle [22]. Nevertheless, in basil plants with intermediary and late flowering, this cultural practice could be eliminated, facilitating handling and reducing costs for the producer.

The mean of the plant heights was of 14.46 cm for the broad leaf accessions and 21.89 cm for those of narrow leaf. In other works [23], a great difference was found in the height of basil plants that varied from 39 to 100 cm, managing to group the six analyzed accessions into three statistically distinct groups in relation to their respective average heights; however, in this work, the accessions presented lower size.

For the diameter of the aerial part and the number of primary branches, the broad leaf accessions presented a mean of 35.80 cm and 5.04, respectively; on the other hand, those of narrow leaf had 44.81 and 7.71 cm, respectively. In works that carried out morphological evaluations of 55 basil genotypes, they also observed great diversity for the crown width variable, which ranged from 18 to 61 cm [24].

The broad leaf accessions presented a mean internode of 0.87 cm, whereas those of narrow leaf presented one of 0.30 cm. Leaf blade width and length, for the broad leaf accessions, were 1.75 and 2.92 cm, and for those of narrow leaf, they were 0.88 and 1.38 cm, respectively (Table 2).

**Table 2.** Plant height (PH), diameter of the aerial part (DAP), number of primary branches (NPB), internode length (IL), leaf blade width (LBW) and length (LBL) for the basil narrow leaf (NL) and broad leaf (BL) accessions.

| Accessions | DAP (cm) | PH (cm) | NPB | IL (cm) | LBW (cm) | LBL (cm) |
|---|---|---|---|---|---|---|
| BL | 35.80 (2.85) [b] | 14.46 (2.97) [b] | 5.04 (1.13) [b] | 0.87 (0.85) [a] | 1.75 (0.17) [a] | 2.92 (0.26) [a] |
| NL | 44.81 (5.56) [a] | 21.89 (3.32) [a] | 7.71 (1.59) [a] | 0.30 (0.26) [b] | 0.88 (0.19) [b] | 1.38 (0.34) [b] |
| C.V. (%) | 11.28 | 14.12 | 21.77 | 34.46 | 15.00 | 16.10 |

Mean (standard deviation). C.V. corresponds to the coefficient of variation. Means followed by the same letter in the columns do not differ from each other at 5% of probability by Tukey's test.

For the percentage of branches with inflorescence, inflorescence branch length, seed maturation and weight, no significant differences were observed among the accessions.

Plant height, diameter of the aerial part and primary branches were inferior in the broad leaf accessions in comparison to the narrow leaves, which presented more compact plants (Figures 1 and 2). This agronomic characteristic is interesting for the cultivation of basil in a hydroponic system aiming at its commercialization *in natura* or for use as an ornamental plant, since a compact plant is produced.

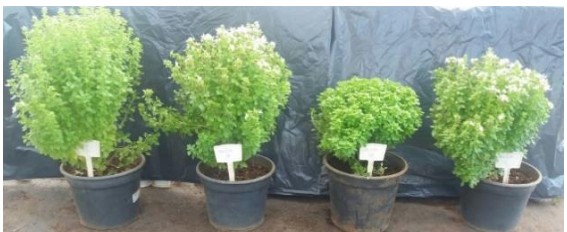

**Figure 1.** Demonstration of the agronomic characteristics of the narrow leaf accessions.

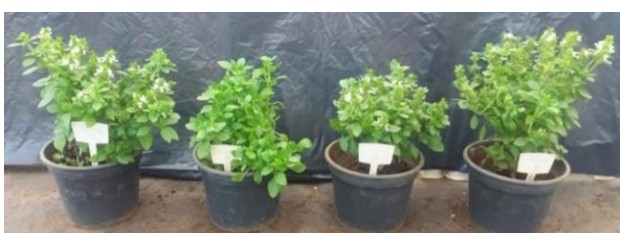

**Figure 2.** Demonstration of the agronomic characteristics of the broad leaf accessions.

Internode length and leaf blade width and length were higher in the broad leaf accessions than in the those with narrow leaf, and the mean of leaf blade length in the broad leaf accession was of 2.92 cm, being higher than in the narrow leaf accession (1.38 cm). A greater internode length, observed in the broad leaf accessions, enables more space for the development of the leaves; therefore, leaves with greater leaf blade width and length were observed in these accessions, whereas for the narrow leaf accessions, the opposite was observed (Figures 1 and 2).

It was verified that plant diameter directly interferes in the quantity and volume of leaves, being an essential characteristic for both the commercialization of the plant *in natura* and for oil extraction. For the extraction of essential oil, plants with a greater number of leaves—that is, greater diameter—are of interest as it will be extracted from these structures [25].

Bigger leaves, such as those of the variety Maria Bonita, which presented leaf length and width of 5.9 and 2.7 cm, respectively, presented high photosynthetic efficiency with photoassimilate storage [26]. The broad leaf accessions with the greatest leaf blade lengths and widths may present this benefit. This characteristic is important from a productivity point of view, both for *in natura* commercialization and for essential oil extraction,

since biomass production is one of the factors that interfere in a greater production of active principles.

A paper [27] mentions some experiments that developed the morphological and agronomic characterization of some basil genotypes [24,28,29], and in these experiments, different crown diameter values were found for different varieties, such as 29.5 cm for Sweet Dani, 34 cm for Genovese, 39.75 cm for Canela and 45.7 cm for Maria Bonita. These values are close to those found in the present work; however, in this one, it is still possible to observe some accessions with higher values, which is interesting when it comes to crown diameter, since it is in this region of the plant where its leaves and flowers are found—the main structures of interest for marketing basil.

*3.2. Second Step: Agronomic Characterization of the Plants in Hydroponic System and Analysis of the Essential Oils*

The accession Manolo presented statistical difference in the diameter of the aerial part, plant height, leaf blade width and length and internode length (Table 3). For the diameter of the aerial part, the other accessions did not present statistical difference from each other. Plant height varied from 0.14 to 0.31 m, with statistical difference among all accessions. This was also observed for leaf blade width and length (0.04 to 1.64 cm and 0.36 to 2.01 cm, respectively). Internode length presented a variation from 0.22 to 6.15 cm among the accessions (Table 3). On the other hand, the fresh mass of the aerial part presented little variation, and the root size and fresh mass presented significant difference among the accessions.

**Table 3.** Diameter of the aerial part (DAP), plant height (PH), leaf blade width (LBW) and length (LBL) and internode length (IL) for the basil narrow leaf (NL) and broad leaf (BL) accessions in hydroponic cultivation 60 days after transplantation.

| Accession | DAP (cm) | PH (cm) | LBW (cm) | LBL (cm) | IL (cm) |
|---|---|---|---|---|---|
| Manolo | 68.08 (1.52) [a] | 31.85 (1.36) [a] | 1.64 (0.04) [a] | 2.01 (0.02) [a] | 6.15 (0.12) [a] |
| NL16 | 48.55(4.00) [b] | 20.83 (0.92) [b,c,d] | 0.58 (0.04) [b,c,d] | 0.92 (0.02) [b,c,d] | 1.92 (0.56) [d,e] |
| BL22 | 47.93 (4.21) [b] | 18.86 (1.15) [c,d,e] | 0.43 (0.07) [d] | 0.76 (0.10) [d,e,f] | 3.02 (0.65) [c] |
| NL19 | 47.77 (0.50) [b] | 21.11 (0.58) [b,c,d] | 0.44 (0.03) [d] | 0.80 (0.01) [d,e] | 2.55 (0.32) [c,d] |
| NL13 | 47.33 (1.66) [b] | 20.77 (0.91) [b,c,d] | 0.51 (0.02) [c,d] | 0.88 (0.02) [b,c,d] | 3.2 (0.35) [b,c] |
| NL38 | 46.08 (1.44) [b] | 15.08 (0.86) [f,g] | 0.04 (0.03) [f] | 0.41 (0.02) [g] | 0.22 (1.25) [e,f] |
| Fino Francês | 46 (4.93) [b] | 22.27 (1.24) [b] | 0.66 (0.07) [b] | 1.00 (0.07) [b,c] | 3.26 (0.34) [b,c] |
| NL28 | 45.4 (3.86) [b] | 20.03 (1.88) [b,c,d] | 0.50 (0.06) [c,d] | 0.86 (0.07) [c,d] | 2.75 (0.63) [c,d] |
| NL35 | 45 (1.86) [b] | 14.25 (0.31) [g] | 0.009 (0.04) [f] | 0.36 (0.07) [g] | 0.84 (0.25) [f] |
| NL26 | 44.46 (2.20) [b] | 21 (1.56) [b,c,d] | 0.24 (0.07) [e] | 0.60 (0.07) [f] | 2.84 (0.18) [c,d] |
| BL24 | 44.13 (4.25) [b] | 16 (1.32) [e,f,g] | 0.65 (0.05) [b,c] | 1.04 (0.03) [b] | 0.32 (3.47) [b,c] |
| NL34 | 43.5 (1.93) [b] | 14.66 (0.47) [g] | 0.01 (0.01) [f] | 0.37 (0.03) [g] | 1.32 (0.19) [e,f] |
| BL11 | 43.22 (2.00) [b] | 18.4 (2.50) [d,e,f] | 0.58 (0.04) [b,c,d] | 0.93 (0.09) [b,c,f] | 4.21 (0.46) [b] |
| NL12 | 40.33 (1.41) [b] | 21 (0.47) [b,c,d] | 0.27 (003) [e] | 0.65 (0.03) [e,f] | 2.41 (0.40) [c,d] |
| NL29 | 39.93 (4.78) [b] | 21.8 (1.29) [b,c] | 0.43 (0.09) [d] | 0.78 (0.07) [d,e] | 3.14 (0.46) [c] |
| C.V. (%) | 7.22 | 6.44 | 11.98 | 7.53 | 14.13 |

Mean (standard deviation). C.V. corresponds to the coefficient of variation. Means followed by the same letter in the columns do not differ from each other at 5% of probability by Tukey's test.

The BL22 and BL11 accessions had a diameter of plant area of 47.93 and 43.22 cm, plant height of 18.86 and 18.4 cm, width and length of the leaf blade of 0.43 and 0.79 cm and 0.58 and 0.93 cm and internode length of 3.02 and 4.21, respectively. These are outstanding agronomic characteristics, demonstrating that these accessions can guarantee a greater production of leaves and a better plant architecture for commercialization and consumption *in natura*. However, future studies can be suggested in order to certify that these characteristics remain highlighted at different times of the year, forms of cultivation and ways of conducting these accessions, in order to validate the adequate way to guarantee the prevalence of these characteristics.

For the construction of the biplot graph (Figure 3), the variable referring to leaf blade width was removed from the database of the principal component analysis in order to solve the problem of multicollinearity since it contributes little to explaining the total data variability.

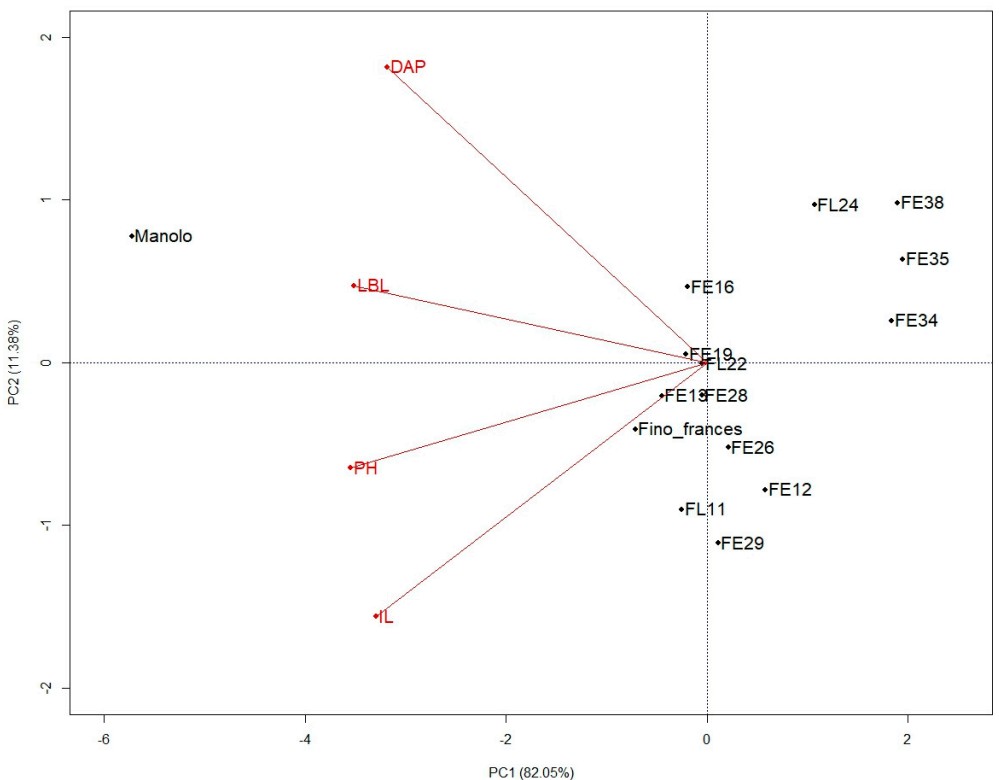

**Figure 3.** Biplot of the variables of diameter of the aerial part (DAP), plant height (PH), leaf blade length (LBL) and internode length (IL) for the basil narrow leaf and broad leaf accessions.

The accession Manolo was prominent among the others in all characteristics observed, presenting the greatest means. Accessions BL11, NL28, NL13, NL19, NL16, NL12, NL26, NL29, BL22 and the cultivar Fino Francês presented some similarities regarding the variables analyzed and were close to the origin, indicating similarity to each other and medium values for the diameter of the aerial part, plant height, leaf blade length and internode length. Conversely, the smallest values were observed for accessions NL34, NL35, NL38 and BL24 (Figure 3).

However, the same author [30] carried out an evaluation of the height of the plants 49 days after transplanting, and he observed that the values were from 0.46 to 0.55 m, also concluding that through studying the development of basil in hydroponics, he noticed a decrease in the development of the same resulting from saline stress. Being able to observe that, in relation to this study, the present work presented a lower average for the height of the plants.

Nevertheless, it is worth highlighting that, although the accession Manolo presents good agronomic characteristics, its cultivation in hydroponics is not interesting, since the plants with greater height and internode are of difficult conduction, falling and generating deformed plants and, consequently, impairing their commercialization.

Another work [30] valuated the height of basil plants in a hydroponic system in different periods of the culture cycle and observed that there was a reduction in this variable as a function of increased water salinity; nevertheless, the values found, on average, were from 21 to 46 cm throughout the assessment days. The values found were close to those obtained in the present work, despite being different basil accessions.

In another work, [9] performed experiments in vases and observed that the basils of the "moita" type were prominent by their average size of 32.3 cm and reduced leaves with leaf area of 4.81 cm$^2$. Nonetheless, in the present work, the thin-leaf accessions observed reached inferior leaf sizes similar to those observed by [9].

In work [9], it was characterized that in the species of *O. basilicum*, the length and width of the leaf blade vary from 3 to 6 cm and from 1.3 to 3 cm. However, as analyzed, among these accessions, only two obtained values within the length, and the others were close, but they were all lower, respectively. This variable can be interesting, because when the leaves are smaller, this can guarantee that a greater number of them are present, being something viable considering that this organ is the most used for commercial purposes of this crop. In addition, this characteristic can attribute advantages to the ornamental use of these plants.

In a protected cultivation of basil in vases, with a 70% shaded system and assessment at 75 days of cultivation, the mean root size was of 18.75 cm, which is a relatively close value to that found in the present work (13.32 cm) [31]. However, it was a characteristic that did not present a significant difference between treatments.

### 3.3. Essential Oil and Sensory Analyses

The estimated contents of essential oil varied from 0.05% to 0.45%, these values being close or even higher than those found by [32] when they analyzed narrow-leaf and broad-leaf basil cultivated in a hydroponic system, since they observed a variation from 0.10% to 0.23%, and by [33], who analyzed 60 different accessions that varied the essential oil content from 0.04 to 0.70%. The accessions with the highest oil contents presented the greatest coloration intensity (high); nevertheless, coloration may be especially related to the compounds present in the oil and not only its total amount in the plant. It is observed that the variable color intensity does not seem to influence the increase or decrease in the variables involved in this analysis. Nevertheless, other studies are necessary to validate this affirmation (Figure 4).

In general, it was possible to observe a grouping between the variables of intensity of odor and eugenol, since they are greatly correlated (r = 0.7539), demonstrating that the greater the odor intensity is, the higher the eugenol content in the accessions is (Figure 4). All accessions that presented the highest levels of eugenol also stood out with the greatest intensity of odor in accessions NL13, NL28 and NL16, which, respectively, presented eugenol contents of 22.7%, 17.95% and 20.95%. On the other hand, accessions NL35, NL38 and NL34 presented the lowest values for these variables (Figure 4). Oil eugenol, another compound also observed in most of the analyzed accessions, may be in the majority together with linalool for some species of basil [34].

In Figure 4, it was also observed that methyl eugenol is inversely correlated (r = 0.6136) to linalool, indicating that accessions with high eugenol content tend to have low linalool content, which corroborates the results obtained, since the highest linalool contents and the lowest contents of methyl eugenol were found in accessions Manolo, BL11, BL24, NL16 and NL28. Analogously inverse, the smallest linalool contents and the highest methyl eugenol contents were obtained for accessions NL26, NL29, NL12, NL34 and BL22 (Figure 4). Conversely, few accessions presented high linalool contents, with highlight only for the variety Manolo (broad leaf) which was not prominent regarding odor. This result may indicate that a greater concentration of linalool does not favor a higher intensity for odor and coloration of these accessions. However, future studies must be carried out to validate this issue.

A prominent accession was NL26 since it presented a methyl eugenol content of 77.7%; then, eugenol was the most abundant component, and finally, linalool was a major component of basil essential oil [12], reaching the highest value of 17.6% in accession BL24. This accession, in turn, was one of the latest for flowering, as observed in the first step of the experiment, starting the reproductive period around seven weeks after transplantation (DAT). It is an accession with very interesting horticultural qualities that can be used as

a new cultivar in programs for the genetic improvement of this condiment, aiming at obtaining new cultivars with high yield of essential oil and commercialization *in natura* because of late flowering.

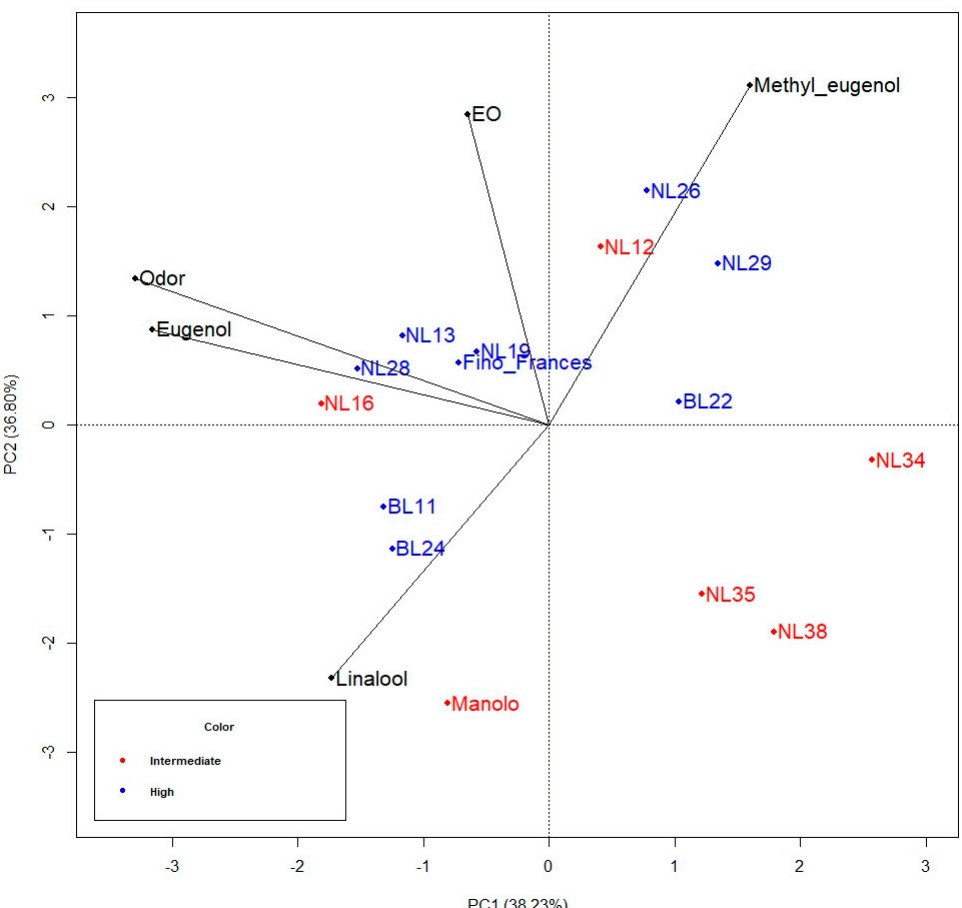

**Figure 4.** Biplot of the variables of estimated content of essential oils (E.O.), color intensity (Color), content of linalool (Linalool), methyl eugenol (Methyl eugenol) and eugenol (Eugenol) and odor intensity (Odor) among the accessions evaluated.

Regarding the content of essential oil, the accessions that presented the lowest extraction values were BL22, Manolo, NL35, NL38 and NL34; on the other hand, the highest values were obtained for accessions NL13, NL28 and NL16 (Figure 4). Therefore, it is important to highlight that information regarding the content and composition of the essential oil (linalool) and late flowering of basil are very important, since it is a condiment commercialized mainly *in natura*. It is thus indispensable that these plants also present a higher number of leaves when commercialized, as well as intense odor to be attractive to the consumer. Therefore, for the selection of the accessions favorable to the market *in natura* and/or for the extraction and commercialization of the essential oil, it is necessary to observe the plants with all these characteristics.

It is important to point out that, in other works [35], the essential oil content was analyzed, obtaining an average of 0.7 to 2.4 mL of essential oil content/100 g of dry forage, and it is important to highlight that, in this work, this analysis was carried out in pre-flowering, which is the stage where the highest levels were found, while the same evaluated the levels at the end of flowering, obtaining lower values for all treatments (0.3 to 1.7 mL 100 $g^{-1}$). This indicates, however, that younger plants with late flowering favor the increase in the essential oil content.

For basil production aiming at essential oil extraction, the highest content is observed with the plant in the post-flowering stage, whereas the lowest content is in the pre-flowering

period [36]. Nonetheless, [37] it was observed that the highest oil content occurred before plant flowering. Thus, the plant development stage can be a determining factor for essential oil content.

Although linalool is the major compound in basil essential oil, as reported by [38], the variation in the composition of the oil can be related to the weather conditions since basils cultivated in Egypt presented as major compounds linalool, methyl chavicol and methyl cinnamate. Nevertheless, more studies are necessary to validate the factors which can influence this variation. In addition, according to [12], other compounds can be found in basil essential oil, due to the wide morphological, biochemical and physiological variation of *O. basilicum* and interspecific hybridization that results in numerous cultivars, thus generating a large difference in volatile fraction contents, which can affect the color and aroma of these plants [39].

Similar to this work, others also found the components linalool, eugenol and methyl chavicol in the composition of basil essential oil [40].

According to [41], methyl eugenol is found in basil leaves, especially when the extraction is performed from dry leaves, as in the present work, which can explain the presence of this compound in all accessions analyzed.

It was observed that there is a variation in the content of essential oil throughout the development of basil plants, with the maximum production occurring before flowering [37]. In this experiment, the evaluated plants were in the vegetative stage (without flowering). This is another point to be considered in the cultivation of basil with late flowering, as analyzed in the first step of this study.

## 4. Conclusions

Considering all accessions evaluated, accessions BL11 and BL22 are highlighted for commercialization *in natura* because of the agronomic characteristics of interest and late flowering. For essential oil production, given its greatest content, BL24 is highlighted. Furthermore, it is important to highlight that in the present work, accessions with superior performance to the varieties used as controls were found, varieties which were inferior in several agronomic characteristics, including essential oil and sensory. Thus, these accessions can be provided as new varieties to be used in programs for the genetic improvement of this species.

**Author Contributions:** Conceptualization, F.A.G.P., M.A.d.S. and F.C.S.; methodology, F.A.G.P., M.A.d.S., S.D.S.d.M. and F.C.S.; software, S.D.S.d.M.; validation, F.A.G.P., M.A.d.S. and F.C.S.; formal analysis, S.D.S.d.M.; investigation, F.A.G.P. and F.C.S.; resources, M.A.d.S., S.D.S.d.M. and F.C.S.; data curation, F.A.G.P. and S.D.S.d.M.; writing—original draft preparation, F.A.G.P., M.A.d.S., S.D.S.d.M., J.M.Q.L. and F.C.S.; writing—review and editing, F.A.G.P., M.A.d.S., S.D.S.d.M., J.M.Q.L. and F.C.S.; visualization, F.A.G.P., M.A.d.S., S.D.S.d.M., J.M.Q.L. and F.C.S.; supervision, F.A.G.P., M.A.d.S., S.D.S.d.M., J.M.Q.L. and F.C.S.; project administration, F.A.G.P. and F.C.S. All authors have read and agreed to the published version of the manuscript.

**Funding:** This research received no external funding.

**Data Availability Statement:** Not applicable.

**Conflicts of Interest:** The authors declare no conflict of interest.

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
