# Peer review of "Agronomic, Sensory and Essential Oil Characterization of Basil (Ocimum basilicum L.) Accessions"

_horticulturae, doi:10.3390/horticulturae9070831_

Round 1

Reviewer 1 Report

Abstract. Include a brief conclusion statement.

Introduction. The novelty of the study is not stated, no previous studies regarding the chemical characterization of basil essential oil were included.

Materials & Methods.

Results. Authors most describe how the identification of the volatile profile was carried out. Why the authors did not use an internal standard? The multivariate analyses (Figures 3 and 4) must include replicates.

Discussion. Authors are poorly integrated and discussed.

Conclusion. OK

English writing is adequate.

Author Response

Dear Reviewer 1

All responses to comments are described in the attached file

Yours sincerely,
Fernanda Abdcuhe.

Reviewer 2 Report

Dear Authors,

The Manuscript ID:  horticulturae-2455228, Titled “Agronomic, sensory and essential oil characterization of Basil (Ocimum basilicum L.) accessions.” as it is written has a number of shortcomings, comments and suggestion.

The title is informative and gives idea of the object of the study.

The Abstract is factual. The aim of the research is clearly stated, but the differences/origin of the 63 basil accessions are not declared and should be written. Additionally, the traits (BL11 and BL24) were not definite. A future perspective should be clearly distinct too.

I should recommend a Graphical abstract to be presented.

The manuscript reports data from a great deal of experimental procedures that appear to be appropriate to the scope of the work. Nevertheless, there is not a significative innovative support to what is widely reported in the literature and it should be supplemented. There is a consistent literature already concerning with the basil cultivars, essential oil characterization and etc. Furthermore, an extra and further experiments are needed.

Discussion. In this section, a good discuss with the results is missing; the data should be interpreted in perspective of previous studies and of the working hypotheses. Future research directions could also be mentioned.

I should recommend Conclusions to be added too.

Finally, based on the evaluation of the manuscript originality, significance of content, scientific soundness, and interest to readers, the manuscript is not appropriate for publication at this stage and further experiments are needed.

Minor editing of English language required

Author Response

Dear reviewer 2

All responses to comments are described in the attached file

Yours sincerely,
Fernanda Abdcuhe.

Reviewer 3 Report

In the article, the authors determined the agronomic characteristics and the content of essential oils in basil plants. A total of 63 basil samples were used.

There are a few comments that need to be corrected to improve the quality of the manuscript:

1. Introduction de provides enough background. A review of similar studies should be made.

2. It is necessary to clearly state the purpose and novelty of the research.

3. It is not entirely clear on what basis the samples of basil were selected?

4. The article lacks a conclusion section. Describe the main results and contributions of this research.

Author Response

Dear reviewer 3

All responses to comments are described in the attached file

Yours sincerely,
Fernanda Abdcuhe.

.

Round 2

Reviewer 1 Report

Authors included all recommendations suggested by the reviewers.

Minor revisions

Author Response

Hello reviewer 1, we would like to thank you for all the considerations and suggestions recommended by you. We hope that we have achieved everything you expected and that we can contribute to the Horticulturae journal.

As for the necessary small editions of the English language, we intend to use the services suggested by the magazine. Let's find out more about them.

Yours sincerely,

Authors.

Reviewer 2 Report

Dear Authors,

You answered to all the comments and suggestions which were given and an enhancement of Manuscript ID: horticulturae-2455228-peer-review-v2 is done.

In conclusion, based on the evaluation of the significance of content, scientific soundness, and interest to readers, I recommend the revised Manuscript to be published in horticulturae.

Author Response

Hello reviewer 2, we would like to thank you immensely for all the considerations and suggestions recommended by you. We hope that we have achieved everything you expected and that we can contribute to the Horticulturae journal.

Yours sincerely,

Authors.

Reviewer 3 Report

The authors corrected all comments in the article. But all the same, the conclusion should be singled out in a separate subsection. I don't have any other questions.

Author Response

Hello reviewer 3, we would like to thank you immensely for all the considerations and suggestions recommended by you. We hope that we have achieved everything you expected and that we can contribute to the Horticulturae journal.

Regarding the conclusion, we highlight it in a separate subsection, as you suggested. We leave this change marked in the submitted manuscript.

Yours sincerely,

Authors.
